# Shifts in phytoplankton community structure across oceanic boundaries

**Jordan Winter**, **Annette Hynes**, **Chris Berthiaume**, **Kelsy Cain**, **E. Virginia Armbrust**, **François Ribalet***

School of Oceanography, University of Washington, Seattle, Washington, United States of America

\* ribalet@uw.edu

**Data availability statement:** Data and code used for the analysis have been deposited in the Zenodo public data repository (DOI: 10.5281/zenodo.14182976).

## Abstract

Phytoplankton communities play an important role in marine food webs and biogeochemical cycles. The transition zones between ocean gyres and surrounding waters represent critical ecological boundaries where environmental gradients drive significant shifts in phytoplankton community structure. This study investigates how nutrient availability and temperature shape the size distribution and composition of small phytoplankton ($< 5$ $\mu$m) communities across the North Pacific Subtropical Gyre (NPSG) boundaries, testing several ecological hypotheses that explain phytoplankton size distribution patterns in relation to environmental variability. We used high-resolution, underway flow cytometry data collected during eight oceanographic cruises from 2016 to 2021 to assess changes in phytoplankton biomass and growth rate across the gyre boundaries. The cyanobacterium *Prochlorococcus* dominated within the gyre, with biomass ranging from 3.2 to 13.1 $\mu$gC L$^{-1}$, and its relative contribution to total phytoplankton biomass varied among cruises (31% to 81%, average 60 $\pm$ 16%). *Prochlorococcus* growth rates were significantly higher within the gyre (0.43 $\pm$ 0.18 per day) than outside the gyre (0.28 $\pm$ 0.16 per day) (one-sided t-test, p $<$ 0.001). Northward in the gyre, *Prochlorococcus* biomass and growth rates declined. Some variations in biomass and growth rates were observed southward and eastward, with biomass ranging from 3 to 10 $\mu$gC L$^{-1}$ and growth rate ranging from 0.2 to 0.6 per day. Outside the NPSG, total phytoplankton biomass increased, with nanoeukaryotes becoming the predominant contributors (up to 71%, 9.1 $\pm$ 7.3 $\mu$gC L$^{-1}$). Picoeukaryote biomass also increased outside the gyre (up to 28 $\pm$ 12% of total biomass). Nutrient concentrations increased by nearly two orders of magnitude outside the NPSG, coinciding with the shift towards larger phytoplankton. The dominance of *Prochlorococcus* within the gyre emphasizes its adaptation to oligotrophic conditions, while the shift towards larger size classes outside the gyre likely reflects the relatively higher nutrient availability. The relatively low abundance of *Synechococcus* even in nutrient-rich regions suggest that that factors beyond nutrient availability, such as grazing, may influence its distribution. These findings have implications for understanding how phytoplankton communities will respond to future changes in oceanographic conditions, such as warming and altered nutrient regimes.

**Funding:** This work was supported by grants from the Simons Foundation (Award IDs #574495 to F.R. and #723795 to E.V.A). There was no additional external funding received for this study.

**Competing interests:** The authors have declared that no competing interests exist.

## Introduction

Phytoplankton are the primary producers of oxygen and organic carbon in marine ecosystems, they form the base of oceanic food webs, and play an essential role in nutrient cycling [1]. Their cell size influences both spatial distributions and functional roles in marine biogeochemical cycles [2,3] through modulation of biotic interactions, including grazing by zooplankton and competition with other phytoplankton, and abiotic interactions, including utilization of different concentrations and forms of nutrients. For example, picophytoplankton, which are less than 2 $\mu$m in diameter, benefit from a high surface area-to-volume ratio that facilitates a more efficient nutrient uptake rate than that of larger cells[2]. This morphological trait is particularly advantageous in nutrient-poor waters, as exemplified by the predominance of the cyanobacterium *Prochlorococcus* (<0.6 $\mu$m) in the oligotrophic subtropical gyres, where it accounts for most of the primary production [2,4,5]. In contrast, larger phytoplankton, categorized as nano- and microphytoplankton (2-20 and 20-200 $\mu$m, respectively), are typically found in regions with higher nutrient availability [6]. Their larger cell size facilitates greater intracellular nutrient storage and a competitive advantage in nutrient-rich environments [2].A shift is apparent from the picocyanobacteria-dominated ecosystem of the North Pacific Subtropical Gyre and a more diverse north subpolar region with larger size classes of phytoplankton [7].

Several hypotheses have been proposed to explain the observed patterns in the size distribution of phytoplankton communities. The "rising tide" hypothesis suggests a relatively uniform response to increasing nutrient availability, where an increase in nutrients leads to a proportional biomass increase across all phytoplankton sizes without altering the size structure of the community [8]. The "step addition" hypothesis posits that under conditions of high, stable nutrient availability, often seen in coastal regions, the biomass of larger phytoplankton increases due to their high nutrient uptake capacity, while the biomass of smaller size classes remains relatively constant [2]. The "enhanced microbial loop" hypothesis proposes that increases in nutrient availability trigger cascading effects throughout the food web, affecting not only phytoplankton but also bacteria, zooplankton and higher trophic levels. The resulting enhanced predation pressure, particularly on smaller cells that are preferentially consumed by grazers, can reshape the size structure of the phytoplankton community [9]. For example, the decline of *Prochlorococcus* with increasing nutrient availability, a pattern attributed to shared grazing pressure with heterotrophic bacteria [7], highlights the intricacies of these interactions. In this scenario, the increased nutrient availability supports not only the growth of larger phytoplankton but also heterotrophic bacteria. The expanding bacterial population, in turn, supports a larger population of grazers that feed on both bacteria and small phytoplankton like *Prochlorococcus*, leading to its decline despite the favorable nutrient conditions. Lastly, the "disturbance hypothesis" highlights how physical disturbances, such as mixing via mesoscale eddies, create a dynamic environment characterized by frequent fluctuations in nutrient availability. These conditions favor smaller, opportunistic phytoplankton, whose faster growth rates and resilience to disturbance-mediated dilution give them a competitive advantage in dynamically disturbed environments [10,11]. These four hypotheses are not mutually exclusive as each offers a different perspective on the relationship between nutrient dynamics and phytoplankton community structure. The change in environmental parameters north, east, and south of the NPSG provides an opportune study site for examining these hypotheses in the context of small phytoplankton. Marine ecosystems are complex and the observed patterns are likely influenced by a combination of these hypotheses, varying in importance depending on the specific environmental conditions. Biotic and abiotic factors

act concurrently, and sometimes antagonistically, to influence the size distribution of phytoplankton communities. This makes it challenging to fully understand the mechanisms driving patterns in phytoplankton size structure [12].

The goal of this study was to elucidate the main environmental drivers underlying the observed patterns in phytoplankton community structure in a region with strong environmental gradients and to evaluate the applicability of various size distribution hypotheses for different groups of small phytoplankton. This study investigates how nutrient availability and temperature shape phytoplankton size distributions across the Northeast Pacific Ocean, encompassing the dynamic transition zone at the boundary of the North Pacific Subtropical Gyre (NPSG) as well as regions north, south and east of the gyre boundary. We collected continuous observations of the abundance and light scatter of small phytoplankton ($< 5 \mu$m) using the shipboard flow cytometer, SeaFlow, during eight oceanographic cruises that covered nearly 40,000 km across regions with different nutrient regimes, from the nutrient-poor NPSG to regions with higher nutrient availability [13]. By analyzing phytoplankton community responses across these environmental transitions, we aimed to test whether the "rising tide," "step addition," "enhanced microbial loop," or "disturbance" hypotheses best explain the observed patterns in different phytoplankton groups and ecological contexts.

## Materials and methods

### Cruises

Eight cruises spanned from the NPSG to surrounding regions (Table 1), transiting to the north, east, and south of the gyre. These cruises ranged in duration from 11 to 31 days, spanning all seasons (spring: March to May, summer: June to August, fall: September to November, winter: December to February) from 2016 to 2021, and covered distances between 3634 km and 7654 km, totaling 39,532 km (Table 1). Each cruise was assigned a season based on the season in which the majority of its cruise days occurred. Cruises were renamed based on direction (first letter) and season (second and third letters) for clarity (e.g., NSU for north summer and EFA for east fall). Note that TN397 transited in multiple directions and was segmented into three distinct transects (EFA1, SFA2 and SFA3) for subsequent analysis. For NSP1, NSP2 and NSP3 cruises, the transect was sampled northward and then southwards, such that the same location was sampled two weeks or so apart.

### Phytoplankton abundance and biomass

Phytoplankton abundance, size and biomass were measured using the underway flow cytometer SeaFlow, which continuously measures forward light scattering (457/50 nm bandpass

**Table 1. Summary of cruises that transit from within to outside the NPSG.**

| cruise | official cruise ID | direction | season | dates | duration (days) | distance (km) |
|---|---|---|---|---|---|---|
| NSP1 | KOK1606 | north | spring | 2016-04-20 to 2016-05-04 | 15 | 3648 |
| NSP2 | MGL1704 | north | spring | 2017-05-26 to 2017-06-13 | 19 | 4566 |
| NSU1 | KM1712 | north | summer | 2017-07-31 to 2017-08-30 | 31 | 5504 |
| NSU2 | KM1713 | north | summer | 2017-09-05 to 2017-09-26 | 22 | 3634 |
| NSP3 | KM1906 | north | spring | 2019-04-10 to 2019-04-29 | 20 | 6263 |
| SFA1 | KM1923 | south | fall | 2019-12-06 to 2019-12-16 | 11 | 3760 |
| EFA1, SFA2, SFA3 | TN397 | east, south | fall | 2021-11-18 to 2021-12-15 | 28 | 7654 |
| EWI1 | TN398 | east | winter | 2021-12-18 to 2021-12-30 | 13 | 4503 |

filter), chlorophyll *a* fluorescence (692/40 nm bandpass filter), and phycoerythrin fluorescence (572/28 nm bandpass filter) of phytoplankton [14]. For seawater collection, we utilized the ship's underway system, which drew samples from depths between 3 and 8 m. The sample water flowed first into an overflow container, then was directed through a filtration step using a 100-$\mu$m stainless steel mesh, which prevented potential blockage of the measurement stream's 200-$\mu$m sampling nozzle (Coulter ML05430). We maintained calibration by continuously introducing 1-$\mu$m yellow-green Fluoresbrite microspheres (Polysciences, Inc.) as reference standards into the flow.

Raw SeaFlow data were first processed as described in Ribalet et al. [15] to identify optimally positioned particles within the SeaFlow virtual core [14]. Subsequent analysis were conducted using the R package popcycle 4.7.3 [15]. We established four distinct classification boundaries using forward light scatter measurements to differentiate between the 1-$\mu$m reference beads, two cyanobacteria groups (*Prochlorococcus* and *Synechococcus*) and small eukaryotic phytoplankton, as described in [16]. Small eukaryotes were further categorized into two populations based on their equivalent spherical cell diameter (see below for details): picoeukaryotes (<2 $\mu$m) and small nanoeukaryotes (2–5 $\mu$m).

To determine cellular dimensions, we calculated equivalent spherical diameter (ESD) from forward light scatter measurements using Mie theory with three refractive indices (1.35, 1.38, 1.41) appropriate for marine phytoplankton [15]. Because scatter intensity can fluctuate with instrument alignment, we selected the optimal refractive index for each cruise by identifying which value produced a median *Prochlorococcus* ESD closest to 0.6 $\mu$m reference measurement established for Station ALOHA [17]. We converted cell diameter measurements to volume (assuming spherical morphology) and subsequently to carbon content using the allometric relationship $Q_C = 0.261 \times V^{0.86}$, where $Q_C$ is the carbon quota (pgC cell$^{-1}$) and V is the volume ($\mu$m$^3$) [18]. Population-specific biomass concentrations were then derived from the product of mean abundance and carbon quota values.

To better capture the spatial variability in biomass, we removed the day-night variability in biomass by applying a classic time series decomposition technique by moving averages [19] to the biomass data. This was achieved using a multiplicative decomposition model with a 24-hour moving window. This allowed us to isolate the trend in biomass and minimize the influence of short-term fluctuations caused by day-night cycles.

Data and code are available in Zenodo (DOI:10.5281/zenodo.14182976).

## Cellular growth rate

Daily cellular growth rates were calculated from daytime increases in carbon quotas, based on the approach described in Hynes et al. [16]. Daytime was calculated from measurements of Photosynthetic Active Radiation (PAR) obtained with each ships' underway light sensor (Biospherical Instruments, San Diego, CA, USA). Specific sensor models ($2\pi$ vs $4\pi$) varied across cruises without detailed documentation, but this variation did not affect our analysis since PAR measurements served only to define daytime periods. To remove spatial variability and better capture daytime changes in carbon quotas, the hourly mean carbon quota ($Q_C$, pg C cell$^{-1}$) data were detrended using the same multiplicative decomposition model with a 24-hour moving window used for biomass data (see above). The daily cellular growth rate ($r$, h$^{-1}$) and the carbon quota at sunrise $Q_{C,0}$ were derived by fitting a transformed exponential growth function to the detrended carbon quota data:

$$\ln(Q_C) = r \times t + \ln(Q_{C,0}), \qquad (1)$$

where $t$ represents the number of hours since sunrise (h), defined as PAR values above 10 $\mu$mol photons m$^{-2}$ s$^{-1}$. The hourly growth rate (r, h$^{-1}$) was converted to daily growth rate (d$^{-1}$) by multiplying by the duration of daylight hours, accounting for the fact that phytoplankton growth primarily occurs during photosynthetically active periods. To estimate cellular growth rate $r$, a given data series had to meet two conditions: 1) estimates were based on more than 6 hourly observations per daytime period (i.e., cell abundance was $\geq 0.02 \times 10^6$ cells L$^{-1}$), and 2) the p-values for the rate estimates were $\leq 0.01$ (indicating the slope of the ln-transformed carbon quota vs. time regression was significantly different from zero) to be considered significant [16].

## Environmental data

Environmental data, including salinity, temperature, and nutrient concentrations, were collected on the different cruises. Salinity and temperature were measured underway using a Seabird Electronics SBE-21 thermosalinograph (Bellevue, WA, USA). Nutrient data were retrieved from datasets publicly available at the Simons Collaborative Marine Atlas Project (Simons CMAP) database [20]. For most cruises, dissolved inorganic nutrients (nitrate, nitrite, and phosphate) were obtained from unfiltered seawater samples. These samples were frozen and analyzed on a SEAL Analytical Autoanalyzer (Mequon, WI, USA). Nitrate and nitrite concentrations below 5 $\mu$mol L$^{-1}$ were analyzed with chemiluminescence [21]. We note that freezing unfiltered seawater may potentially cause cell lysis, which could release intracellular nutrients into solution. However, given the relatively low phytoplankton biomass in our study regions (typically 5-50 $\mu$gC L$^{-1}$)[22], we estimate using Redfield stoichiometry that complete cell lysis would contribute at most 8-30 nmol L$^{-1}$ of phosphate and 100-500 nmol L$^{-1}$ of nitrogen. These potential contributions are generally small compared to the ambient nutrient concentrations measured and would not significantly impact our comparative analyses across different oceanic regions.

Measured nutrient data values came from the following CMAP datasets: tblKOK1606_ Gradients1_Nutrients, tblMGL1704_Gradients2_Nutrients, tblKM1906_Gradients3_Organic _Inorganic_Nutrients, tblTN397_Gradients4_NutrientsAndParticulates, tblGradients5_ TN412_NutrientsUW, tblTN398_nutr for measured nutrient values. For NSU1, NSU2 and SFA1 cruises, we used PISCES modeled nutrient data from Mercator_Pisces_Biogeochemistry _Daily_Forecast_Nut.

## Distance from the gyre boundary

The salinity front was used to estimate the gyre boundary and was defined as the maximum absolute derivative of salinity with respect to distance along each cruise track [7]. To smooth high-frequency variations and improve the accuracy of the derivative calculation, a spline was fitted to the salinity values prior to the differentiation. In cases where salinity changed multiple times along a cruise track, the salinity front closest to the average NPSG salinity of 35 PSU [7] was selected as the gyre boundary. The gyre boundary was designated as 0 km distance. Negative distances indicate locations inside the gyre (calculated as the distance from the salinity front), and positive distances represent locations outside the gyre (also calculated as the distance from the salinity front). The transition zone, as defined by the region of rapid salinity changes, was on average 30 km wide ($\pm$ 30 SD).

## Data analysis

The high-frequency raw data were aggregated and binned over a 24-hour period to minimize temporal autocorrelation. Given an average ship speed of 10 knots, 24-hr represents data

binned over an average distance of 450 km. For each 24-hour bin, the mean value was calculated for each variable (phytoplankton biomass for each group, growth rates, temperature, salinity, dissolved inorganic nitrogen, and dissolved inorganic phosphorus).

A Welch Two Sample t-test was performed to test the difference between the daily binned means inside and outside the gyre (level of significance of 0.01) for temperature and salinity.

A Pearson correlation analysis was performed between phytoplankton biomass, growth rates, and environmental factors to identify statistically significant linear relationships (level of significance of 0.01) and quantify the strength of these associations. The correlation matrix included pairwise correlations between all phytoplankton groups (*Prochlorococcus*, *Synechococcus*, picoeukaryotes, and nanoeukaryotes), their biomass, growth rates, and the environmental variables (temperature, salinity, and concentrations of dissolved inorganic nitrogen and phosphorus). Missing values were handled by casewise deletion (i.e. if a specific bin had a missing value for either variable in a correlation pair, data from that bin was deleted from the analysis), so the correlation between each pair of variables was computed using all complete pairs of observations on those variables. This resulted in a sample size ranging from 73 to 132 observations, with a average of 118 observations per correlation pair. A Benjamini & Hochberg adjustment was applied to the p-values from the correlation analysis to control the false discovery rate at 0.01 [23].

All our statistical analysis was conducted in R (version 3.6.2). The code is available in Zenodo (DOI:https://doi.org/10.5281/zenodo.14182976).

## Results

The eight analyzed cruises (Table 1) traveled within both the North Pacific Subtropical Gyre (NPSG) and along transits to the north (NSP1, NSP2, NSU1, NSU2, NSP3), east (EFA1, EWI1), or south (SFA1, SFA2 and FA3) of the NPSG (Fig 1a). For each cruise, distinct salinity fronts were detected at approximately 32 °N on the northern edge, between 5 and 7 °N on the southern edge, and between 27 and 29 °N on the eastern edge (Fig 1b), which were used to define gyre boundaries [7]. In the NPSG, sea surface temperatures ranged from 16 °C to 29 °C, with an average of 25 °C across all cruises (Fig 1c). Within the northern and eastern regions of the NPSG, salinity and sea surface temperature generally decreased with increasing proximity to the gyre boundary. In the northern region (NSP1, NSU1, NSU2, NSP3), salinity decreased from an average of 34.8 PSU to 34.1 PSU, and sea surface temperature decreased from an average of 26.3 °C to 20.6 °C. Similarly, in the eastern region (EFA1, EWI1), salinity decreased from an average of 34.8 PSU to 34.1 PSU, while sea surface temperature decreased from 26.3 °C to 20.6 °C (Fig 1c, 1d). In contrast, within the southern region of the NPSG (SFA1, SFA2 and SFA3), salinity increased from 33.5 PSU to 35.1 PSU with increasing proximity to the gyre boundary and were significantly lower than outside the gyre (one-sided t-test, $p < 0.001$, N = 266 and 268 inside and outside the gyre, respectively), while sea surface temperatures remained relatively constant, averaging around 25.8 °C (one-sided t-test, p = 0.03, N = 266 and 268 inside and outside the gyre, respectively). Additionally, the average temperatures of the NPSG were lower during the winter and spring cruises (EWI1, NSP1, NSP2, NSP3) (24.2 °C) than in summer and fall cruises (NSU1, NSU2, SFA1, EFA1a,b,c and EWI1) (25.4 °C) (one-sided t-test, $p < 0.001$).

Dissolved inorganic macronutrient concentrations increased by nearly two orders of magnitude along the northern and southern transects outside the NPSG. Dissolved inorganic nitrogen (DIN) concentrations increased from below 0.05 $\mu$mol L$^{-1}$ within the gyre to almost 11 $\mu$mol L$^{-1}$ outside and dissolved inorganic phosphate (DIP) increased from below 0.2 $\mu$mol L$^{-1}$ within the gyre to 1.1 $\mu$mol L$^{-1}$ at the northernmost station (Fig 1e, 1f).

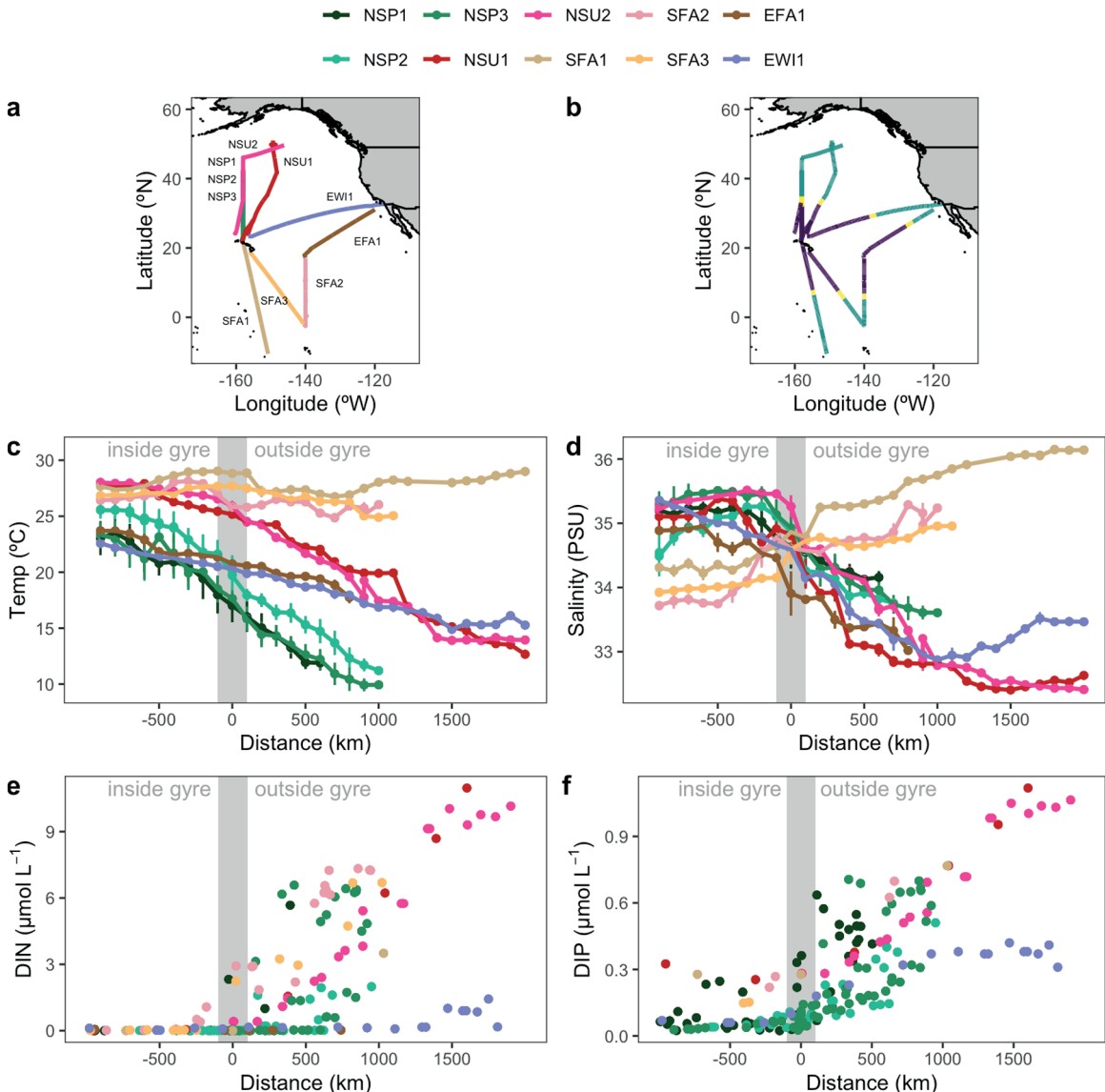

**Fig 1. Environmental gradients and nutrient distributions across distance from the gyre boundary.** (a) Cruise tracks, with each colored line representing a different cruise, with some cruises overlapping in certain regions (i.e., NSP1, dark green, NSP2, green, NSP3, light green, and NSU2, pink, in the north). (b) Salinity (PSU) and (c) temperature (°C) along each cruise track, plotted as a function of distance (km) from the gyre boundary. (d) Each cruise was divided into three regions based on change in salinity (see Methods): the NPSG (purple), the gyre boundary (yellow), and the region outside the NPSG (teal). (e) Dissolved inorganic nitrogen (DIN) and (f) dissolved inorganic phosphorus (DIP) concentrations ($\mu$mol L$^{-1}$) along each cruise track. Negative distances indicate locations inside the gyre, while positive distances represent locations outside the gyre. along each cruise track. Error bars in c) and d) represent standard deviations of the binned data over 100 km intervals (average N = 10 per bin). The gray shaded area highlights the gyre boundary.

Similar increases in DIN and DIP were observed along the southern transects. In contrast, the two eastward cruises displayed increases in DIP but not in DIN, with DIN remaining low at greater than 1500 km outside the gyre during the EWI1 cruise.

Total biomass of the <5 $\mu$m phytoplankton was consistently higher outside the NPSG to the north than within the NSPG, averaging across the nortward cruises 19.7 $\pm$ 12.7 $\mu$gC L$^{-1}$

outside and $11.1 \pm 7.0 \,\mu$gC L$^{-1}$ within (Fig 2). For each phytoplankton group, changes in biomass were primarily driven by changes in cell abundance (Fig 2b), as cell size showed little variation inside and outside the gyre (Fig 3). Within the NPSG, the measured biomass was dominated by the cyanobacteria *Prochlorococcus*, with smaller proportional contributions by *Synechococcus*. With increasing distance outside the northern or southern gyre edges, the measured biomass became dominated by the small eukaryotic phytoplankton. In contrast, the relative proportion of different <5 $\mu$m phytoplankton types remained relatively constant with distance within about 1000 km from the eastern edge of the NPSG.

*Prochlorococcus* carbon biomass within the NPSG ranged from 3.2 to 13.1 $\mu$gC L$^{-1}$ (Fig 2) and consistently dominated the total biomass of the small phytoplankton across all seasons

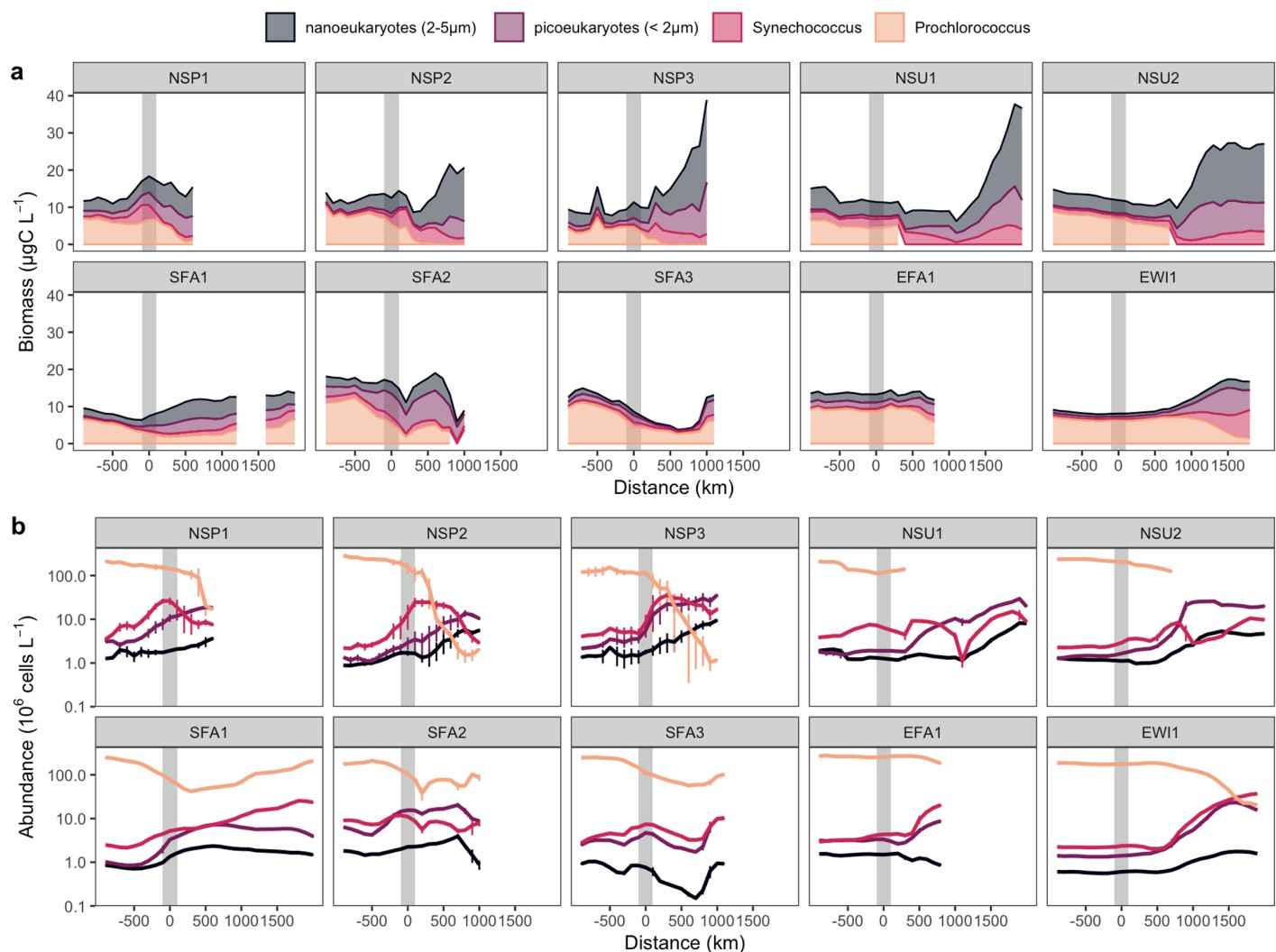

**Fig 2. Phytoplankton biomass and cell abundance across distance from the gyre boundary** a) Phytoplankton biomass ($\mu$g C L$^{-1}$) and b) cell abundance ($10^6$ cells L$^{-1}$) relative to distance (km) from the gyre boundary (grey bar), with negative distances indicating locations within the gyre and positive distances representing locations outside the gyre. Colored lines represent the biomass of different phytoplankton groups: *Prochlorococcus* (orange), *Synechococcus* (red), picoeukaryotes (<2 $\mu$m, purple), nanoeukaryotes (2-5 $\mu$m, black). *Prochlorococcus* abundance drops below detection during NSU1 and NSU2 outside the gyre. Error bars in b) represent standard deviations of the binned data over 100 km intervals (N = 1-7 per bin, average N = 2). The gray shaded area highlights the gyre boundary.

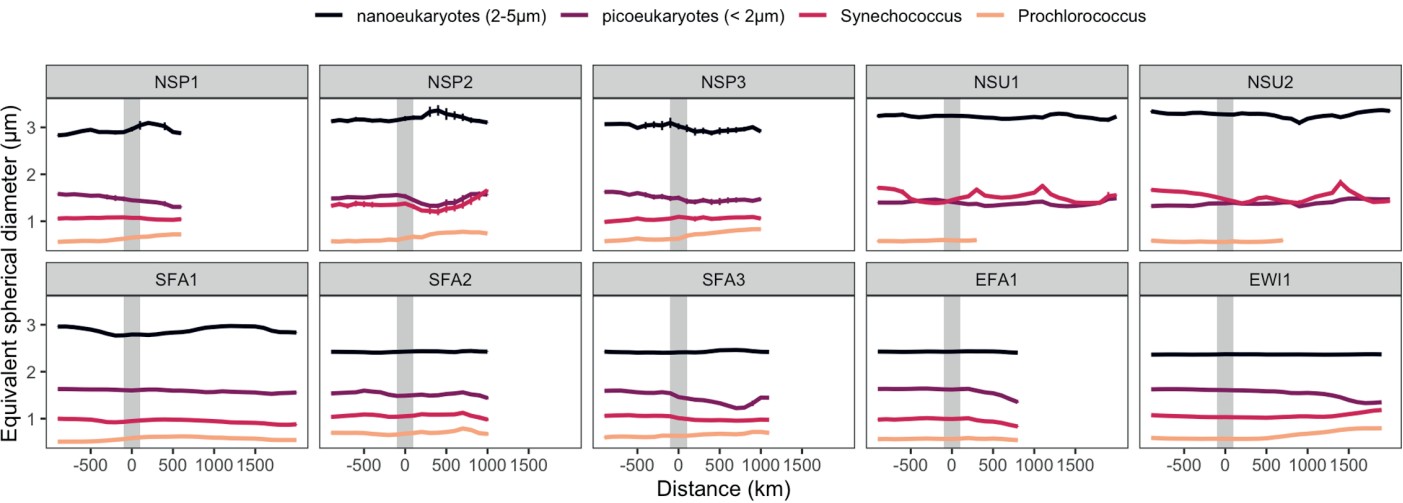

**Fig 3. Phytoplankton cell size across distance from the gyre boundary.** Phytoplankton cell size ($\mu$m ESD, equivalent spherical diameter) plotted against distance (km) from the gyre boundary (grey bar), with negative distances indicating locations within the gyre and positive distances representing locations outside the gyre. Colored lines represent the biomass of different phytoplankton groups: *Prochlorococcus* (orange), *Synechococcus* (red), picoeukaryotes (<2 $\mu$m, purple), nanoeukaryotes (2-5 $\mu$m, black). Error bars represent standard deviations of the binned data over 100 km intervals.

and cruise directions. However, its relative contribution to the total small phytoplankton biomass varied among cruises and ranged from 31% to 81%, with an average of 60% $\pm$ 16% (7.7 $\pm$ 2.0 $\mu$gC L$^{-1}$), with a notable 3-fold difference between the two southbound fall cruises (SFA1: 3.5 $\pm$ 1.5 $\mu$gC L$^{-1}$; SFA3: 9.7 $\pm$ 1.7 $\mu$gC L$^{-1}$). This variation in biomass (Fig 2a) was mirrored by changes in cell abundance (Fig 2b), which ranged from 2.9 $\times$ 10$^8$ to 0.9 $\times$ 10$^5$ cells L$^{-1}$ within the NPSG. North of the gyre boundary, *Prochlorococcus* biomass declined sharply, dropping from the within-gyre average of 7.2 $\mu$g C L$^{-1}$ to less than 1 $\mu$g C L$^{-1}$ within 500 km. Southward, the decline in *Prochlorococcus* biomass was more gradual, with an initial decrease followed by a slight increase towards the southernmost points, except for SFA2 where it continuously declined. East of the gyre, *Prochlorococcus* biomass remained relatively stable at $\sim$3 $\mu$g C L$^{-1}$ before declining near the easternmost points, corresponding to the California coast (at 750 km and 1500 km east of the boundary for EFA1 and EWI1, respectively).

*Synechococcus* biomass was generally low within the NPSG across all cruises (1.2 $\pm$ 0.9 $\mu$g C L$^{-1}$), consistent with the low cell abundance within the NPSG (4.9 $\times$ 10$^6$ $\pm$ 3.8 $\times$ 10$^6$ cells L$^{-1}$) (Fig 2b). Northward outside the gyre, *Synechococcus* abundance increased approximately 3 fold at the salinity front in spring (NSP1, NSP2 and NSP3), reaching 18 $\times$ 10$^6$ cells L$^{-1}$ and contributing up to 47% (10.7 $\mu$gC L$^{-1}$) of the total phytoplankton biomass (Fig 2). This pattern was less pronounced to the east, with *Synechococcus* biomass and cell abundance remaining stable or slightly increasing within 1000 km of the gyre boundary. No such increase at the salinity front south of the gyre was observed.

Picoeukaryote biomass was low within the NPSG and increased outside the gyre across all cruises, with the most substantial increase observed in the northern region (Fig 2a). On the northern cruises, biomass increased from an average of 1.0 $\pm$ 0.6 $\mu$g C L$^{-1}$ inside the gyre to 6.4 $\pm$ 4.0 $\mu$g C L$^{-1}$ outside, while on the eastern cruises, the increase was more moderate (from 1.3 $\pm$ 0.5 $\mu$g C L$^{-1}$ to 3.0 $\pm$ 2.3 $\mu$g C L$^{-1}$). This increase in biomass across all cruises was accompanied by a corresponding rise in cell abundance, which increased from an average of 2.7 $\times$ 10$^6$ cells L$^{-1}$ inside the gyre to 14.9 $\times$ 10$^6$ cells L$^{-1}$ outside (Fig 2b).

Nanoeukaryotes also exhibited higher biomass outside the gyre, particularly in the north, where they became the predominant contributors of phytoplankton <5 $\mu$m (up to 71%), with an average biomass of 9.1 $\pm$ 7.3 $\mu$gC L$^{-1}$ and a maximum value of 27.4 $\mu$gC L$^{-1}$. This was mirrored by an increase in cell abundance from 1.3 $\times$ 10$^6$ cells mL$^{-1}$ within the gyre to 3.4 $\times$ 10$^6$ cells mL$^{-1}$ outside the gyre (Fig 2b). In contrast, nanoeukaryote biomass and abundance remained relatively stable in the southern region within the gyre (1.9 $\pm$ 0.6 $\mu$gC L$^{-1}$ and 1.2 $\times$ 10$^6$ $\pm$ 0.4 $\times$ 10$^6$ cells mL$^{-1}$) and gradually increased towards the boundary in the eastern cruises.

We used the change in per cell carbon quota over the daily cycle to estimate cellular net growth rates [16]. *Prochlorococcus* growth rates were generally higher within the gyre (0.43 $\pm$ 0.18 per day) compared to outside (0.28 $\pm$ 0.16 per day) (one-sided t-test, p < 0.001) (Fig 4). This pattern was particularly evident in two of the northern cruises (NSP1 and NSP2), where growth rates decreased outside the gyre as the cruises progressed northward into cooler waters, aligning with the observed decline in *Prochlorococcus* biomass (Fig 2a). However, *Prochlorococcus* growth rates was highly variable outside the gyre among other cruises, with some showing relatively constant net growth rates (EWI1) or even increased net growth rates outside the gyre (NSU2). *Synechococcus* exhibited significantly lower overall net growth rates (0.24 $\pm$ 0.11 per day) than *Prochlorococcus* (0.36 $\pm$ 0.18 per day) (one-sided t-test, p < 0.001). *Synechococcus* growth rates were similar inside (0.26 $\pm$ 0.12 per day) and outside (0.23 $\pm$ 0.11 per day) the gyre (two-sided t-test, p = 0.2), with no clear trends related to cruise direction or location relative to the gyre boundary (Fig 4). Picoeukaryote net growth rates were not significantly different outside the gyre (0.18 $\pm$ 0.09 per day) compared to inside (0.20 $\pm$ 0.11 per day) (two-sided t-test, p = 0.03). Nanoeukaryotes displayed significantly lower net growth rates (0.11 $\pm$ 0.07 per day) than picoeukaryotes (0.19 $\pm$ 0.11 per day) (two-sided t-test, p < 0.001). Both picoeukaryote and nanoeukaryote growth rates were highly variable, with no clear pattern discernible in relation to the gyre boundary or cruise direction (Fig 4).

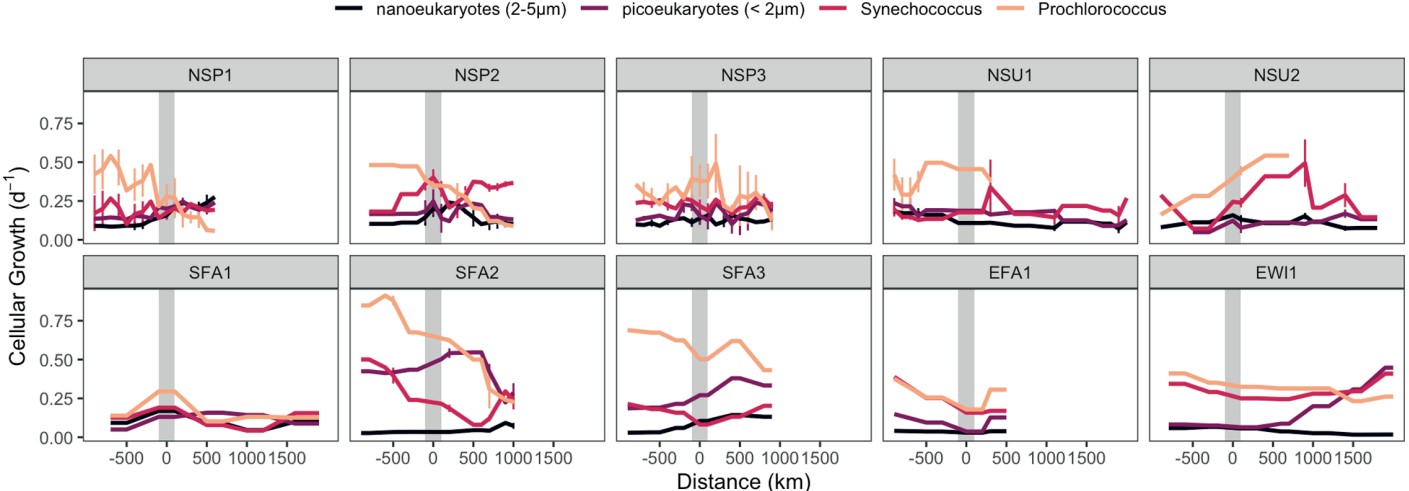

**Fig 4. Phytoplankton cellular growth rates across distance from the gyre boundary.** Phytoplankton cellular growth rates (d$^{-1}$) relative to distance (km) from the gyre boundary (grey bar), with negative distances indicating locations within the gyre and positive distances representing locations outside the gyre. Colored lines represent the cellular growth rates of different phytoplankton groups: *Prochlorococcus* (orange), *Synechococcus* (red), picoeukaryotes (<2 $\mu$m, purple), nanoeukaryotes (2-5 $\mu$m, black). *Prochlorococcus* abundance drops below detection during NSU1 and NSU2 outside the gyre. Error bars represent standard deviations of the binned data over 100 km intervals. Note that error bars are shown when the ship spent more than 24 hours within a 100 km interval.

To further explore the factors influencing variations in growth rate and biomass, we examined the correlations between these phytoplankton measures with key environmental parameters, including temperature and nutrient concentrations. *Prochlorococcus* growth rate was positively correlated with temperature (Pearson r = 0.60, p < 0.001, N = 114) and negatively correlated with DIP (Pearson r = -0.39, p < 0.001, N = 73) (Fig 5). In contrast, the growth rates of *Synechococcus*, picoeukaryotes, and nanoeukaryotes showed no significant correlations with temperature or nutrient concentrations (p > 0.06 for temperature, and p > 0.4 for nutrients). A weak yet significant correlation was observed between the growth rates of picoeukaryotes and nanoeukaryotes with that of *Prochlorococcus* (Pearson r = 0.1 and 0.3, respectively, p < 0.01 and N = 114 for both). While the growth rates of *Synechococcus*, picoeukaryotes, and nanoeukaryotes appeared largely independent of the environmental factors examined, their biomass showed strong negative correlations with temperature (Pearson r = -0.52, -0.71 and -0.70, respectively, p < 0.001 and N = 132 for all). Furthermore, picoeukaryote and nanoeukaryote biomass showed positive correlations with both DIN (Pearson r = 0.55, p < 0.001 and r = 0.66 and p < 0.001, respectively, N = 85 for both) and DIP (r = 0.48, p < 0.001 and r = 0.56, p < 0.001, respectively, N = 73). Finally, biotic interactions

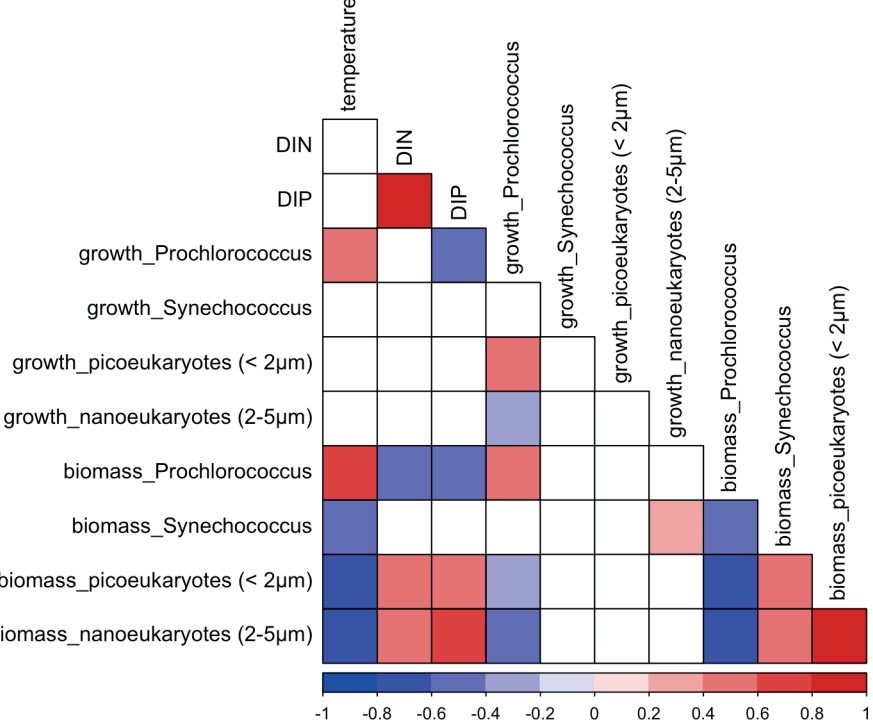

**Fig 5. Nutrient-phytoplankton relationships.** Correlation matrix between temperature, dissolved inorganic nitrogen (DIN) and dissolved inorganic phosphorus (DIP), cellular growth rate of *Prochlorococcus*, *Synechococcus*, picoeukaryotes (<2 $\mu$m), nanoeukaryotes (2-5 $\mu$m), and their biomass. Only correlations with statistical significance (p < 0.01, after Benjamini & Hochberg correction for multiple comparisons) are shown; positive correlations are in shades of red, negative correlations in blue, with color intensity indicating the strength of the Pearson correlation coefficient (r). White cells indicate non-significant correlations (p > 0.01). Sample size varied for different correlation pairs due to casewise deletion of missing values (n ranged from 73 to 132 observations, with a average of 118 observations per correlation pair).

likely further contributed to the observed patterns in biomass distribution. *Prochlorococcus* biomass were negatively correlated with *Synechococcus*, picoeukaryote and nanoeukaryote biomass (Pearson r = -0.48, -0.67, and -0.74, respectively, p < 0.001 and N = 110 for all), and picoeukaryote biomass was positively correlated with nanoeukaryote biomass (r = 0.73, p < 0.001, N = 128).

## Discussion

Our results demonstrate that the boundary of the NPSG is a critical region where dramatic shifts in phytoplankton community structure and size distribution occur. These shifts appear to be driven by a complex interplay of factors, including nutrient availability and temperature. The dominance of *Prochlorococcus* within the gyre across different seasons and cruise directions, highlights its remarkable adaptability to the oligotrophic conditions characteristic of this region. Its small size, high surface area-to-volume ratio, and efficient nutrient uptake mechanisms confer a competitive advantage in nutrient-poor waters, supporting the long-standing evidence of its significance in such environments [2,4,5]. However, the observed variability in *Prochlorococcus* biomass within the gyre suggest that factors beyond nutrient limitation influence its abundance. Temperature, in particular, emerged as a strong positive correlate of *Prochlorococcus* growth (r = 0.60) and biomass (r = 0.67) (Fig 5), suggesting its potential sensitivity to thermal regimes within the gyre. This variability contrasts with the more stable patterns observed at Station ALOHA, a long-term time-series station located north of Hawaii that has been regularly sampled since 1988 [5,16,24]. While Station ALOHA provides valuable temporal data at a fixed location within the NPSG, our spatial survey highlights the dynamic and heterogeneous nature of the gyre environment that encompasses such a large area of the North Pacific, with a 13°C change in temperature and a 2 PSU variation in salinity observed within the gyre [1,25].

The contrasting patterns observed in *Prochlorococcus* biomass and growth rates between northern and southern cruises underscore the influence of temperature on its distribution. The decline in biomass and growth rates in northward cruises, coinciding with decreasing temperatures, suggests a temperature-dependent limitation on its physiology, directly linking its physiological activity with population dynamics [26,27]. This sensitivity to lower temperature, coupled with potential negative biotic interactions such as shared grazing pressure with heterotrophic bacteria [7] and shared viral lysis with *Synechococcus* [28], may contribute to the observed decline of *Prochlorococcus* populations in the northern regions, in agreement with ecosystem model predictions [12]. In these cooler northern waters, competitive dynamics also shift as larger phytoplankton groups gain advantages in nutrient acquisition. Nanoplankton, rather than microplankton, biomass has also been found to significantly increase to the north of the gyre, likely due to size-specific grazing and the increase in nutrient concentrations [22]. The physiological constraints imposed by lower temperatures on *Prochlorococcus* may reduce its competitive ability against these larger phytoplankton, which are better adapted to cooler, more nutrient-rich conditions. In contrast, the relatively stable biomass and growth rates observed in the warmer waters of the southern cruises, highlight *Prochlorococcus*' preference for higher temperatures. This further supports the growing body of evidence that temperature plays a crucial role in shaping the distribution and activity of *Prochlorococcus* in the ocean [26,27,29]. The favorable growth temperatures in the south and east of the gyre likely facilitate the persistence of *Prochlorococcus* populations [29]. Additionally, the advection of nutrients from the equatorial Pacific [30] and potential iron limitation [31] in these regions may prevent larger phytoplankton from outcompeting *Prochlorococcus* [32].

While we observed an increase in the total biomass of phytoplankton with increasing nutrient availability, this increase was not uniform across all size classes as predicted by the "rising tide" hypothesis [8]. Instead, we observed a shift in the size distribution towards larger phytoplankton, particularly nanoeukaryotes, indicating that relatively higher nutrient availability favors the growth of larger cells over smaller ones. This pattern aligns with the "step addition" hypothesis, which predicts an increase in the biomass of larger phytoplankton in response to greater nutrient availability [2].The ability of larger cells to store nutrients and capitalize on resource pulses likely contributes to their success in these more eutrophic regions. Changes in nutrient concentrations on a seasonal cycle may also influence whether nanoplankton or microplankton are more successful in the midlatitudes [3]. The positive correlations between nanoeukaryote biomass and both DIN and DIP ($r = 0.61$ and $r = 0.38$, respectively) support the notion that nutrient availability drives their growth, with nutrient availability limiting their growth in the gyre. The dominance of nanoeukaryotes outside the NPSG, especially in northern cruises, emphasizes their ecological significance in these nutrient-rich environments [22]. While the positive correlation between picoeukaryote and nanoeukaryote biomass ($r = 0.69$) suggests a degree of co-occurrence or shared environmental preferences for these two groups, *Prochlorococcus* biomass declined substantially outside the gyre. The negative correlations between *Prochlorococcus* biomass and both picoeukaryote and nanoeukaryote biomass ($r = -0.70$ and $r = -0.67$, respectively) hint at resource partitioning among these size classes or potential competitive interactions. For instance, the decline of *Prochlorococcus* populations at the northern boundary of the NPSG has been attributed to increased grazing pressure and viral infections [12,28,33]. These nuances highlight the complex interplay between biotic and abiotic factors in shaping community structure.

While nanoeukaryotes thrive in less oligotrophic conditions, smaller phytoplankton groups like picoeukaryotes and *Synechococcus*, exhibit more complex responses to relatively higher nutrient availability, potentially due to limitations in nutrient competition and increased grazing pressure. Picoeukaryotes, for instance, may face limitations in nutrient competition due to their smaller size and lower nutrient storage capacity compared to nanoeukaryotes. Furthermore, the increase of nanoeukaryotes may also lead to an increase in "shared predators" that preferentially feed on smaller phytoplankton, keeping picoeukaryotes in check despite the increased nutrient availability, consistent with the "enhanced microbial loop" hypothesis [9]. *Synechococcus* presents an intriguing case in our study. Despite increased nutrient availability outside the gyre, its biomass and growth rates remained relatively low throughout most sampled regions. However, we observed localized increases in *Synechococcus* biomass near the salinity front, particularly in the northern and eastern boundaries. This pattern suggests a unique ecological niche for *Synechococcus* at these transition zones. The boundary regions likely experience heightened physical dynamics including enhanced mixing and fluctuating nutrient inputs [34]. These conditions align with the ecological concept that intermediate disturbance can create opportunities for opportunistic taxa. *Synechococcus*, with its intermediate size and adaptable physiology, appears positioned to exploit these transitional environments where neither the larger eukaryotes nor *Prochlorococcus* dominate completely. The reduced predation pressure at boundary zones may also contribute to this pattern, as the abundance of carnivorous predators often peaks at these frontal systems [35], potentially disrupting the grazing chain that would otherwise control *Synechococcus* populations. This spatial distribution suggests that *Synechococcus* may function as an ecological indicator of boundary dynamics, occupying a specialized niche in regions characterized by environmental variability rather than stable conditions of either the oligotrophic gyre interior or the more consistently nutrient-rich waters beyond. Understanding

these distribution patterns has important implications for predicting how shifts in oceanic boundaries due to climate change might reshape phytoplankton community structure in the future.

## Conclusions

Our findings indicate a combination of the "step addition" hypothesis, "enhanced microbial loop" hypothesis, and "disturbance" hypothesis reflect the community composition changes in and around the NPSG. The "step addition" hypothesis is supported by the increase in larger phytoplankton in response to greater nutrient supply outside the gyre. The "enhanced microbial loop" hypothesis is supported by the complex responses of smaller phytoplankton to higher nutrient availability, potentially due to increased grazing pressure. The "disturbance" hypothesis is supported by the localized increase in *Synechococcus* biomass at the gyre boundary, potentially due to reduced grazing pressure and nutrient variability. However, our findings do not support the "rising tide" hypothesis, as the increase in total phytoplankton biomass with increasing nutrient availability was not uniform across all size classes.

## Acknowledgments

We thank Katherine Qi for her insights on estimating cellular growth rates using detrended carbon quota data.

## Author contributions

**Conceptualization:** E. Virginia Armbrust, Francois Ribalet.

**Data curation:** Jordan Winter, Annette Hynes, Chris Berthiaume, Kelsy Cain.

**Formal analysis:** Jordan Winter, Kelsy Cain.

**Funding acquisition:** E. Virginia Armbrust, Francois Ribalet.

**Investigation:** Jordan Winter, E. Virginia Armbrust, Francois Ribalet.

**Methodology:** Jordan Winter, Annette Hynes, Chris Berthiaume, Kelsy Cain, Francois Ribalet.

**Project administration:** Francois Ribalet.

**Resources:** Francois Ribalet.

**Software:** Annette Hynes, Chris Berthiaume, Francois Ribalet.

**Supervision:** E. Virginia Armbrust, Francois Ribalet.

**Validation:** Annette Hynes, Chris Berthiaume.

**Visualization:** Jordan Winter.

**Writing – original draft:** Jordan Winter.

**Writing – review & editing:** Jordan Winter, E. Virginia Armbrust, Francois Ribalet.

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
