## [Decision Letter · Decision Letter 0]

PONE-D-24-52976Environmental Gradients Drive Shifts in Phytoplankton Community StructurePLOS ONE

Dear Dr. Ribalet,

Thank you for submitting your manuscript to PLOS ONE. After careful consideration, we feel that it has merit but does not fully meet PLOS ONE’s publication criteria as it currently stands. Therefore, we invite you to submit a revised version of the manuscript that addresses the points raised during the review process.

**In particular, reviewers have concerns about the lack of clarity in the way the data is presented (and make suggestions for improvement) which makes the paper difficult to follow. There is a concern about plagiarism with the authors previous works. This is a serious matter and requires your attention. A careful review of the manuscript will improve its readability and access to a wider audience.**

We look forward to receiving your revised manuscript.

Kind regards,

Antonietta Quigg, PhD

Academic Editor

PLOS ONE

**Journal Requirements:**

This work was supported by grants from the Simons Foundation (Award IDs #574495 to F.R. and #723795 to E.V.A).

We thank Katherine Qi for her insights on estimating cellular growth rates from detrended carbon quota data. This work was supported by grants from the Simons Foundation (Award IDs #574495 to F.R. and #723795 to E.V.A).

This work was supported by grants from the Simons Foundation (Award IDs #574495 to F.R. and #723795 to E.V.A).

4. We notice that your supplementary figures are included in the manuscript file. Please remove them and upload them with the file type 'Supporting Information'. Please ensure that each Supporting Information file has a legend listed in the manuscript after the references list.

**Additional Editor Comments:**

This study investigates the dynamics of small phytoplankton (< 5 μm) across the North Pacific Subtropical Gyre (NPSG), using high-resolution, underway flow cytometry data collected during eight oceanographic cruises from 2016 to 2021. The authors conclude that the dominance of Prochlorococcus within the gyre emphasizes its adaptation to oligotrophic conditions, while the shift towards larger size classes outside the gyre likely reflects nutrient enrichment. The relatively low abundance of Synechococcus even in nutrient-rich regions suggest that that factors beyond nutrient availability, such as grazing, may influence its distribution.

I ran into several challenges with reading the paper as summarize them below. i ask the authors to consider these and revise the manuscript to improve the readability of the paper. There are also specific comments below.

(1) in the introduction, the authors present four possible hypotheses to explain their observations. in the discussion, the authors fail to contextualize their findings within these frameworks. While they discuss two of the four hypotheses as possible explanations for their observations, they do not wrap around full circle to explain why the two other hypotheses are a no go.

(2) when describing the nutrient data, the authors fail to provide the reader with tangible information to understand how the nutrient data relates to the plankton data. Where the nutrient data collected hourly? every x km? some other variable? given the plankton data was binned into pool of 100km, how many nutrient samples does that reflect? 1, 10. 20? more information is needed than just pointing the reader to a map.

(3) when describing the stats, the authors fail to mention how they handled any missing data. Given they examined 8 cruises over many years, there surely is missing data. how did the authors address this issue?

(4) ** methods which caused issues in results and discussion ** i had a tremendously difficult time trying to keep track of which cruise corresponded with which direction, which season, then you have all sorts of colors in figure 1, etc... i recommend the reviewers consider a naming convention for the paper e.g., KOK1606 could be NSP1 = north spring 1... KM1712 could be NSU1 = north summer 1... this would help the reader when looking at your figures and reading the text. i got very frustrated trying to align what was in the figures with what was in the text.

(5) Fig. 1 d Each cruise was divided into three regions based on change in salinity (see Methods): the NPSG (purple), the gyre boundary (yellow), and the region outside the NPSG (teal)

- I couldn’t work this out from methods provided. Pls give more details

(6) Please consider making Figure 1 into two figures with Fig 1a and 1 d as a new figure 1 and figures b,c,e and f as a second figure. this will help the reader see the error bards more easily - especially as it looks like there are no error bars on some of the data in this figure.. is that because there was only one sample or is this because the error bars are so small, you cant see them. given you didnt explain the specifics for the nutrient data, the reader doesnt know why this is missing... i am also surprised that the error bars get so small on temp and salinity for some but not all cruises... whats going on?

(7) method for making figure 4 missing from methods section, please add

(8) Line 123 you talk about Photosynthetic Active Radiation (PAR) measurements but no write up on these is given in the paper. was this information just used to determine day/night? if yes, where their distinct patterns between day and night? we know large phytoplankton vary; this would be a nice opportunity to know if small phytoplankton vary in the time they spend in surface waters. In the discussion, you say complex factors affect distribution... can you rule out that day versus night is not one of those factors for these groups?

(9) the discussion on the disturbance hypothesis at the end of the discussion doesnt align with the hypothesis itself. please rephrase this section

Line 155 to Line 158 - the text and the figure do not match, please fix

Line 158 – please check the temperature values in this paragraph. They do not match what I see in fig. 1c. salinities appear reasonable (fig. 1b) but salinities are definitely not matching.

Line 290 with a 10°C change in temperature and a 2 PSU variation in salinity observed across the gyre ??? the temperature and PSU changes were certainly more dramatic - look at figure 1

Reviewers' comments:

Reviewer's Responses to Questions

**Comments to the Author**

1. Is the manuscript technically sound, and do the data support the conclusions?

Reviewer #1: Yes

2. Has the statistical analysis been performed appropriately and rigorously? 

Reviewer #1: Yes

3. Have the authors made all data underlying the findings in their manuscript fully available?

Reviewer #1: Yes

4. Is the manuscript presented in an intelligible fashion and written in standard English?

Reviewer #1: Yes

5. Review Comments to the Author

**Reviewer #1:** PONE-D-24-52976 Environmental Gradients Drive Shifts in Phytoplankton Community Structure

Review, 13 January 2025.

The manuscript presents collected data to examine marine cyanobacterial and small eukaryotic phytoplankton species across spatial gradients from within to outside a subtropical gyre within the northern Pacific Ocean. The data are collected from on-board flow cytometry during a number of research cruises, and analyzed to present clear gradients in relative abundance and biomass between size-related phytoplankton groups, in relation to physical-chemical parameters to chart the gyre boundaries. The data are generally well described and presented and the ‘story’ about the size distribution changes within and outside the gyre is interesting and are nicely discussed in the context of previously published hypotheses driving biotic interactions and abiotic constraints on phytoplankton populations. The relatively brief manuscript presents valuable data and perspectives and should be published. However, some details of methodology including replication and nutrient measurement, need to be clarified, and several details need to checked and reviewed for accuracy. Although the journal criteria do not specifically require strong rationale, strengthening the overall rationale would help the reader follow the ideas and purpose of the study (and ultimately help the paper receive more attention and thus citations). This material is all present in the manuscript but just requires some clearer statement of the aims/objective. Detailed suggestions to help improve the manuscript and specific points to address in the revision are listed below.

Abstract

The rationale for examining the populations of picoplankton within and outside of the gyre is well described in the Introduction with clear theoretical background, but this rationale and motivation is missing from the Abstract. After first two very general background sentences of the abstract, it launches into what was done, but it needs some statement of the question, problem, hypothesis, aim or idea that is motivating this study. And then what follows, “This study investigates the dynamics of small phytoplankton” is rather vague. Even for a publication in PLOSone, the abstract would benefit from a clearer idea of what the study was aiming to achieve.

Abstract (and line 332) “…while the shift towards larger size classes outside the gyre likely reflects nutrient enrichment.” Nutrient enrichments suggests and injection of nutrients from some source. The open ocean outside the gyre is not really nutrient ‘enriched’ so maybe express as ‘relatively higher nutrient availability’ or ‘less nutrient depleted’ or similar.

Introduction

Line 18-49. The authors present a nice array of hypotheses to explain the size distributions and follow-up on these dominates the Discussion, but linking these to what the study aims to achieve seems to be missing. The goal listed later (line 57-) could incorporate the analysis of these data to examine these potential hypotheses including different drivers of populations.

Line 49. Something about spatial gradients to preface what is going to be examined in the paper? (…the word gradient in is the title but not mentioned anywhere else in the text except the figure legends)

Line 50 – a statement of the aim or objective would be useful before what was done… Promote (and edit) the material in line 57-60 to have a more up-front goal or rationale, rather than as a post-script to what was done.

Line 33 and … “nutrient supply…” – measured standing concentration value of nutrients, not a supply rate. So nutrient availability to cells can relate to the concentration, and the rate of turnover from different pools. The authors should consider this distinction in their descriptions and interpretations.

Materials and Methods

Some text description is very similar (or identical) to previously published methods (e.g. lines 75-81, 82-86, 90-98, are very similar to Hynes et al. 2024). Although the methods are clearly the same, to avoid self-plagiarism, the authors should modify their text and/or cite the work and avoid repeating segments of text from previous papers.

On line 96, 97, the authors define carbon quota as Qc,0 but later this is defined as the Qc at sunrise (line 114). In this section (Phytoplankton Abundance and Biomass), the calculation of Qc from cell diameter makes sense, but not that this is Qc,0 as no time is defined. The Qc,0 defined as at sunrise later (line 114) makes sense in section ‘Cellular Growth Rate’, but not on line 96-97. However, if the inputs to this calculation are cell C content, then is this really a cell-based growth rate or a biomass-based growth rate? (see also figure 3). In figure 3, the authors also introduce ‘net’ in the y axis scale (this is the first appearance of ‘net’ in the manuscript). This should be defined and consistent terminology used throughout the manuscript text and figures.

Line 119 – what statistical tests were used to determine the significance of the rate estimates? What was compared to derive these significance values?

Line 124. 2 pi or 4 pi sensor? Model used?

Line 125. If the water samples were not filtered then frozen then any cells in the sample could have been lysed, contributing some unassimilated nutrients to the analyses, depending on the analytical techniques applied in the AutoAnalyzer function. It might be useful to have some commentary about likely contribution of these cell nutrient sources and description of the methods used so the reader can evaluate contribution of nutrients other than orthophosphate and ammonium/ammonia. Line 20 has a confusing post-script suggesting that the nutrient data came from another source? The authors need to clarify whether they analyzed the data in their study or used data from a database?

Line 145-6. What factors were used in the ANOVA for e.g. to determine if growth rate estimates were significant (see line 119)? A bit more statistical data would be helpful but this could be provided in the results when the specific comparison results are presented, and/or on the figure legends.

Results

Line 152-155. This sentence is grammatically incomplete.

Line 158-165. Are these changes spatially, or differences across the gyre significantly different (see lines 167-172 where differences are compared statistically. How many temperature values are being averaged in these stated average values, and over what time periods?

Line 173, 236, 244, 249 etc – t-test is not mentioned in the M&M section (lines 138-149).

Line 180. To check and understand this point, it is hard to see Tn397a on plots in Figure 1e, 1f – are there data missing or just under other points. Lines 181-183 seem like Discussion points of interpretation?

Line 184-186 – are these averages cited from all cruises and all timepoints?

Lines 186-188. This is a valuable and important point for interpretation of growth of phytoplankton cells, so not sure if the journal restricts figure numbers, but I would inclined to include this figure rather than making it supplementary, so the reader has them all in one place!

Line 210, 219-225. Are these averages and variation across all samples and all transects and time?

Line 226. Nanoeukaryotes also exhibited higher biomass outside the gyre.

Line 227 (and elsewhere). The authors state ‘predominant contributors…’.[to populations?] outside the gyre. Is this just within the picoplankton size class examined, or also when microplankton (e.g. diatoms etc) are considered? This context is not clear.

Figures

Figure 1F. Not sure what the 10X in the y axis label refers to? Based on the numbers, I do no think this means that the values on the plot are 10 times than the scale bar on the y axis??

Figure 1 legend “Error bars represent standard deviations of the binned data over 100 km intervals”. It would be useful to include a range of data values these represent i.e. N = range?

It might be useful just in figure 1B to include ‘inside gyre’ and ‘outside gyre’ inside at the top of the plot to help the reader, to support the legend description of negative and positive distances.

Figure 2. Are these data plotted against environmental gradients (as the legend suggests) or solely against location (distance)? The legend title is a bit misleading. As biomass is calculated from cell number and cell size, and cell size did not change much spatially (Figure S1), the plots at the top and the bottom show similar trends, but the abundance shows particularly that Prochlorococcus is numerically dominant in the gyre, if not always in terms of biomass. “Error bars represent standard deviations of the binned data over 100 km intervals.” – as for figure 1, include the range of N (error bars are evident in some plots but not others, so this clarification would be useful - clarified in Fig. 3 only, but context of data replication of measurements would be valuable).

Fig 3. Phytoplankton Cellular Net Growth Rates Across Environmental Gradients … see comment on figure 2. The x axis is distance, not any environmental parameter. The y axis terminology is also inconsistent with the M&M (see comment above). This is also different from the data in Figure 4 of Hynes et al. (2024) which was cited for the calculation method. In that paper, the growth rate range (~ 0.05 h^-1 for Prochlorococcus or 1.2 d^-1) is very different to those growth rates reported here (max ~ 0.05 d^-1 for Prochlorococcus within the gyre). So it makes me wonder if the calculations are the same, or are showing something different? Or if the ‘net’ means that something has been subtracted, but this is not mentioned in the M&M.

Figure 3 legend end. “DIN concentrations (μmol L−1) are shown as black dots” – I cannot see any black dots in or on the plots nor do I see any scale for DIN concentration on any axis.

Fig. 2 and 3. Color between pico and nanoeuks is a bit hard to differentiate when the lines are close.

Figure 4 – white is not shown on the scale – is that indicating no significant correlation, or not determined?

Discussion

Line 289. It may be useful to provide context for the comparison with Station ALOHA which is also in the NPSG – how these would one expect this difference to be explained? The reader may not have a detailed understanding of the areas sampled for this study vs ‘more stable patterns’ at Station ALOHA so it may be useful to explain this.

Line 295-296. The authors may provide some more competitive context here (see comment above about microplankton), along with the viruses and grazing effects. The effects of competition are mentioned (line 307-8) in the southern region, but what about in northern regions where temperature becomes more limiting for Prochlorococcus?

Line 310 (some perspective on even larger size classes (microplankton) here would be valuable.

Line 317. “…nutrient availability drives their growth.” I think we know that nutrients drive growth but what this suggests is that nutrient availability limits the growth of the nanoEuks (within the gyre).

Line 331, 333, 341, 364. ‘relatively higher nutrient availability’ rather than nutrient enriched conditions

6. PLOS authors have the option to publish the peer review history of their article (what does this mean?). If published, this will include your full peer review and any attached files.

Reviewer #1: No

---

## [Author Response · Author response to Decision Letter 1]

23 Apr 2025

This study investigates the dynamics of small phytoplankton (< 5 μm) across the North Pacific Subtropical Gyre (NPSG), using high-resolution, underway flow cytometry data collected during eight oceanographic cruises from 2016 to 2021. The authors conclude that the dominance of Prochlorococcus within the gyre emphasizes its adaptation to oligotrophic conditions, while the shift towards larger size classes outside the gyre likely reflects nutrient enrichment. The relatively low abundance of Synechococcus even in nutrient-rich regions suggest that that factors beyond nutrient availability, such as grazing, may influence its distribution.

I ran into several challenges with reading the paper as summarize them below. I ask the authors to consider these and revise the manuscript to improve the readability of the paper. There are also specific comments below.

1. in the introduction, the authors present four possible hypotheses to explain their observations. in the discussion, the authors fail to contextualize their findings within these frameworks. While they discuss two of the four hypotheses as possible explanations for their observations, they do not wrap around full circle to explain why the two other hypotheses are a no go.

We have added information in the Discussion (L374-378) to address the hypotheses presented in the Introduction, and the Conclusions (L429-439) has been entirely restructured to articulate how the combination of 'step addition,' 'enhanced microbial loop,' and 'disturbance' hypotheses together provide the most comprehensive explanation for the observed community shifts across the NPSG boundary.

2. when describing the nutrient data, the authors fail to provide the reader with tangible information to understand how the nutrient data relates to the plankton data. Where the nutrient data collected hourly? every x km? some other variable? given the plankton data was binned into pool of 100km, how many nutrient samples does that reflect? 1, 10. 20? more information is needed than just pointing the reader to a map.

We have redesigned the presentation of nutrient data in Figure 1. Instead of showing mean values across 100 km bins, we now display all individual nutrient data points, allowing readers to see the actual sampling density and variability. Additionally, we have clarified in the Methods (L145-167) that nutrient data were obtained from publicly available repositories through Simons CMAP database, with appropriate dataset accession information now included.

3. when describing the stats, the authors fail to mention how they handled any missing data. Given they examined 8 cruises over many years, there surely is missing data. how did the authors address this issue?

Missing values were handled by casewise deletion, so if a specific location has an NA for any of the variables involved in the correlation calculation, data from that location is deleted from the analysis. We added that information in the Methods L197-201.

4. ** methods which caused issues in results and discussion ** i had a tremendously difficult time trying to keep track of which cruise corresponded with which direction, which season, then you have all sorts of colors in figure 1, etc... i recommend the reviewers consider a naming convention for the paper e.g., KOK1606 could be NSP1 = north spring 1... KM1712 could be NSU1 = north summer 1... this would help the reader when looking at your figures and reading the text. i got very frustrated trying to align what was in the figures with what was in the text.

We implemented a consistent naming convention for all cruises throughout the manuscript, with new designations reflecting both direction and season (e.g., NPS1 for North Pacific Spring cruise 1), described L77-78. Table 1 now includes these updated names alongside the original cruise identifiers. The color scheme in Figure 1 has been revised to visually distinguish both season and direction. We added a note about this naming convention in the Methods section for clarity.

5. Fig. 1 d Each cruise was divided into three regions based on change in salinity (see Methods): the NPSG (purple), the gyre boundary (yellow), and the region outside the NPSG (teal) - I couldn’t work this out from methods provided. Pls give more details

We added to the section “distance from the gyre boundary” L175-179 to clarify the definition of the three regimes given in fig. 1.

6. Please consider making Figure 1 into two figures with Fig 1a and 1 d as a new figure 1 and figures b,c,e and f as a second figure. this will help the reader see the error bards more easily - especially as it looks like there are no error bars on some of the data in this figure.. is that because there was only one sample or is this because the error bars are so small, you cant see them. given you didnt explain the specifics for the nutrient data, the reader doesnt know why this is missing... i am also surprised that the error bars get so small on temp and salinity for some but not all cruises... whats going on?

We restructured Figure 1 into three distinct rows to improve the visibility of error bars and data points. Regarding the larger standard deviations observed in temperatures and salinity for NPS1, NPS2, and NPS3 cruises, we have added an explanation in the Methods (L80-82) clarifying that these locations were sampled twice approximately two weeks apart, resulting in natural temporal variability in environmental conditions

7. method for making figure 4 missing from methods section, please add

We have opted not to include specific details about the creation of figure 4 in the manuscript due to space constraints and the complexity of the figure generation process. However, to ensure reproducibility, we have made the code and documentation for generating all figures available in Zenodo (DOI:10.5281/zenodo.14182976).

8. Line 123 you talk about Photosynthetic Active Radiation (PAR) measurements but no write up on these is given in the paper. was this information just used to determine day/night? if yes, where their distinct patterns between day and night? we know large phytoplankton vary; this would be a nice opportunity to know if small phytoplankton vary in the time they spend in surface waters. In the discussion, you say complex factors affect distribution... can you rule out that day versus night is not one of those factors for these groups?

Thank you for this thoughtful suggestion about exploring diel patterns. While diel patterns influence phytoplankton distribution (a topic we discussed previously in Ribalet et al. 2015 and Hynes et al, 2024, both cited in our manuscript), this analysis falls outside the scope of our current study, which focuses primarily on spatial gradients and nutrient influences across the NPSG boundary. We mentioned in the Methods (L115-120) that we detrended the biomass data over 24-hour periods specifically to minimize the impact of diel cycles on our analysis of broader spatial patterns. This approach allowed us to isolate the environmental gradient effects that were central to our research questions.

9. the discussion on the disturbance hypothesis at the end of the discussion doesnt align with the hypothesis itself. please rephrase this section

We restructured this section to discuss in more details the intriguing distribution of Synechococcus observed in our study, L408-427.

Line 155 to Line 158 - the text and the figure do not match, please fix

Corrected.

Line 158 – please check the temperature values in this paragraph. They do not match what I see in fig. 1c. salinities appear reasonable (fig. 1b) but salinities are definitely not matching.

Corrected.

Line 290 with a 10°C change in temperature and a 2 PSU variation in salinity observed across the gyre ??? the temperature and PSU changes were certainly more dramatic - look at figure 1

Corrected.

Comments to the Author

5. Review Comments to the Author

Reviewer #1: PONE-D-24-52976 Environmental Gradients Drive Shifts in Phytoplankton Community Structure

Review, 13 January 2025.

The manuscript presents collected data to examine marine cyanobacterial and small eukaryotic phytoplankton species across spatial gradients from within to outside a subtropical gyre within the northern Pacific Ocean. The data are collected from on-board flow cytometry during a number of research cruises, and analyzed to present clear gradients in relative abundance and biomass between size-related phytoplankton groups, in relation to physical-chemical parameters to chart the gyre boundaries. The data are generally well described and presented and the ‘story’ about the size distribution changes within and outside the gyre is interesting and are nicely discussed in the context of previously published hypotheses driving biotic interactions and abiotic constraints on phytoplankton populations. The relatively brief manuscript presents valuable data and perspectives and should be published. However, some details of methodology including replication and nutrient measurement, need to be clarified, and several details need to checked and reviewed for accuracy. Although the journal criteria do not specifically require strong rationale, strengthening the overall rationale would help the reader follow the ideas and purpose of the study (and ultimately help the paper receive more attention and thus citations). This material is all present in the manuscript but just requires some clearer statement of the aims/objective. Detailed suggestions to help improve the manuscript and specific points to address in the revision are listed below.

Abstract

1. The rationale for examining the populations of picoplankton within and outside of the gyre is well described in the Introduction with clear theoretical background, but this rationale and motivation is missing from the Abstract. After first two very general background sentences of the abstract, it launches into what was done, but it needs some statement of the question, problem, hypothesis, aim or idea that is motivating this study. And then what follows, “This study investigates the dynamics of small phytoplankton” is rather vague. Even for a publication in PLOSone, the abstract would benefit from a clearer idea of what the study was aiming to achieve.

We have revised the abstract introduction to explicitly state our research questions and theoretical framework. The new opening establishes why understanding picoplankton community shifts across gyre boundaries is important and how our study contributes to testing established hypotheses about phytoplankton size distribution in oligotrophic versus mesotrophic waters.

2. Abstract (and line 332) “…while the shift towards larger size classes outside the gyre likely reflects nutrient enrichment.” Nutrient enrichments suggests and injection of nutrients from some source. The open ocean outside the gyre is not really nutrient ‘enriched’ so maybe express as ‘relatively higher nutrient availability’ or ‘less nutrient depleted’ or similar.

We changed both of these instances to “relatively higher” nutrient availability. Also changed all other instances of the use of the word “enrichment” in this context to avoid this confusion.

Introduction

1. Line 18-49. The authors present a nice array of hypotheses to explain the size distributions and follow-up on these dominates the Discussion, but linking these to what the study aims to achieve seems to be missing. The goal listed later (line 57-) could incorporate the analysis of these data to examine these potential hypotheses including different drivers of populations.

We restructured the Introduction to establish a stronger connection between the presented hypotheses and our study objectives. We moved the research rationale to the beginning of the paragraph (L54-57) and explicitly framed our study as a test of these competing hypotheses (L65-68). This restructuring creates a clearer narrative flow from theoretical framework to specific research aims.

2. Line 49. Something about spatial gradients to preface what is going to be examined in the paper? (…the word gradient in is the title but not mentioned anywhere else in the text except the figure legends)

We changed the title to “Shifts in Phytoplankton Community Structure Across Oceanic Boundaries”

3. Line 50 – a statement of the aim or objective would be useful before what was done… Promote (and edit) the material in line 57-60 to have a more up-front goal or rationale, rather than as a post-script to what was done.

Moved the statement to the start of the paragraph, L54-57.

4. Line 33 and … “nutrient supply…” – measured standing concentration value of nutrients, not a supply rate. So nutrient availability to cells can relate to the concentration, and the rate of turnover from different pools. The authors should consider this distinction in their descriptions and interpretations.

Changed “supply” to “availability” throughout

Materials and Methods

1. Some text description is very similar (or identical) to previously published methods (e.g. lines 75-81, 82-86, 90-98, are very similar to Hynes et al. 2024). Although the methods are clearly the same, to avoid self-plagiarism, the authors should modify their text and/or cite the work and avoid repeating segments of text from previous papers.

We have completely rewritten these methodology sections using distinct phrasing. Where appropriate, we now cite Hynes et al. (2024) to acknowledge methodological similarities and avoid duplication of the text.

2. On line 96, 97, the authors define carbon quota as Qc,0 but later this is defined as the Qc at sunrise (line 114). In this section (Phytoplankton Abundance and Biomass), the calculation of Qc from cell diameter makes sense, but not that this is Qc,0 as no time is defined. The Qc,0 defined as at sunrise later (line 114) makes sense in section ‘Cellular Growth Rate’, but not on line 96-97. However, if the inputs to this calculation are cell C content, then is this really a cell-based growth rate or a biomass-based growth rate? (see also figure 3). In figure 3, the authors also introduce ‘net’ in the y axis scale (this is the first appearance of ‘net’ in the manuscript). This should be defined and consistent terminology used throughout the manuscript text and figures.

We have dropped the “,0” subscript when referring to the general carbon quota estimated form cell diameter, at this is not specifically tied to sunrise measurements.

Since we are using the change in per-cell carbon content over time, our method is clearly a cellular-level measurement, not a cell abundance or population biomass measurement.

The "net" qualifier was meant to acknowledges that the measured rate represents the balance between carbon fixation and any losses (respiration and exudation), but we recognize that it may add confusion with “net growth rate” commonly used to describe the net change in population growth rate over time. We have dropped to “Net” in the y-axis label of the figure.

3. Line 119 – what statistical tests were used to determine the significance of the rate estimates? What was compared to derive these significance values?

The p-values was derived from the regression analysis - specifically testing whether the slope coefficient (r) is significantly different from zero - the null hypothesis being that there is no growth (r = 0). We added that information in the Methods L142-143 and in the caption of Figure 4.

4. Line 124. 2 pi or 4 pi sensor? Model used?

The PAR sensors varied across the eight oceanographic cruises, and we did not maintain a detailed record of the specific sensor models (2π or 4π) used on each vessel. However, this does not affect our analysis since we only used PAR measurements to define daytime periods (values above 10 µmol photons m⁻²s⁻¹) for growth rate calculations, rather than for any quantitative modeling. The absolute PAR values and sensor geometry were not critical for this binary day/night determination. We added that information in the Methods L124-129.

5. Line 125. If the water samples were not filtered then frozen then any cells in the sample could have been lysed, contributing some unassimilated nutrients to the analyses, depending on the analytical techniques appli

---

## [Editor Report · Decision Letter 1]

Shifts in Phytoplankton Community Structure Across Oceanic Boundaries

PONE-D-24-52976R1

Dear Dr. Ribalet,

We’re pleased to inform you that your manuscript has been judged scientifically suitable for publication and will be formally accepted for publication once it meets all outstanding technical requirements.

Kind regards,

Antonietta Quigg, PhD

Academic Editor

PLOS ONE
---

## [Editor Report · Acceptance letter]

PONE-D-24-52976R1

PLOS ONE

Dear Dr. Ribalet,

I'm pleased to inform you that your manuscript has been deemed suitable for publication in PLOS ONE. Congratulations! Your manuscript is now being handed over to our production team.

Kind regards,

on behalf of

Dr. Antonietta Quigg

Academic Editor

PLOS ONE